## Article
# Seed Morphology and Germination of Native *Tulipa* Species

**Wei Zhang [1,2], Jun Zhao [2], Li Xue [2], Hanping Dai [2] and Jiajun Lei [2,*]**

1   College of Horticulture, Shanxi Agricultural University, Taigu 030801, China
2   College of Horticulture, Shenyang Agricultural University, Shenyang 110866, China
*   Correspondence: jiajunlei@syau.edu.cn; Tel.: +86-024-88487143; Fax: +86-024-88487146

**Abstract:** Seeds are important for the preservation and reproduction of wild tulips in situ, but only a few studies have been carried out on their seed morphology and germination characteristics. In this study, the eight wild tulip species native to China were used. Their seed morphology and superficial ornamentation, the relationship between seed morphology and germination, and the effects of manipulating incubation temperature, seed coat, and gibberellin ($GA_3$) application were studied. The seeds are diverse in shape, size, color, weight, and superficial ornamentation, as observed by stereomicroscopy and scanning electron microscopy. *Tulipa edulis* (Miq.) Baker had a half-moon-shaped seed with the embryo invisible and irregular striped ornamentation, which was different from the other seven species that had sector-shaped seeds with the embryo visible and irregular reticulate ornamentation. Especially, the muri feature and the decorative pattern in meshes were highly variable and decisive at both section and species levels of *Tulipa* L. We also classified the seed dormancy type of the eight wild tulip species as nondeep complex (MPD). Embryo length, embryo/seed length ratio, and seed coat color were correlated with germination ($T_{50}$), while the $T_{50}$ of eight wild tulip species was still mainly affected by optimized temperatures of 4 °C and/or 4/16 °C. Gibberellin ($GA_3$) has a positive regulating effect on the $T_{50}$ of tulip species, and 100 μM gibberellin was considered the most effective concentration. These results highlight the taxonomic significance of the genus *Tulipa* based on seed testa morphology, and we support the notion that *T. edulis* should be regarded as an independent genus—*Amana*. Meanwhile, our study provided a way that the seed germination of wild tulip species could be shortened within 2 months under the experimental conditions, which will help to rapidly multiply and utilize wild tulip resources.

**Keywords:** *Tulipa*; seed morphology; germination; temperature; hormone

## 1. Introduction

The tulip, in the genus *Tulipa* of the family Liliaceae, is a perennial bulbous plant that is grown in pots and gardens for bedding, ground cover, and cutting [1,2]. It is known as 'the Queen of Bulb Flowers' for its flower colors and shapes. The Tianshan and Pamir Altai Mountains in Central Asia were believed to be the primary gene centers for the *Tulipa* genus [3–5]. The genus *Tulipa* includes approximately 100 species, and is distributed naturally in southern Europe, North Africa, the Middle East, and Central Asia, including Western China [6,7]. In China, there are 13 recognized species in this genus, and the majority of them are distributed mainly in the Xinjiang Province [8–10]. In recent years, these wild resources have drawn many researchers' attention, in particular to their cultivation, domestication, and reproductive biology [11,12]. These wild resources have some useful ornamental characteristics and strong adaptability to the environment, which can be used as important resources to breed new tulip cultivars.

Seeds are more resistant to the poor environment than at any other stage in the plant life cycle, and sexual reproduction plays an important role in the spread and establishment of many species [13–15]. Recent studies have also shown that seed coats can furnish important information for determining taxonomic relationships in many plants [16–18]. The study of seed morphology is helpful in understanding the survival strategies and evolution

trends of plants in fragmented habitats. Seed size is a relatively stable character, which is an important feature of plant life history [19]. Seed weight of a plant species is generally considered to be a relatively stable character that affects its germination and seedling establishment [20,21]. Tan [9] believed that the ultrastructure of seed coat patterns observed by scanning electron microscope (SEM) could be considered reliable and informative for elaborating phylogenetic and taxonomic hypotheses of *Tulipa*. In addition, seed size is an important physical indicator of seed quality, which can affect vegetative growth [22]. Morphological measurements, including length, width, and weight, are important parameters for determining the size and shape of seeds [23]. Seed shape and size can influence water imbibition, seed moisture content, and consequently seed germination [24,25]. For tulips, a lot of research has been focused on its seed morphology, but few studies have been reported on the relationship between seed morphology and germination characteristics in the genus *Tulipa* in detail.

Tulip seed has the characteristics of dormancy, which is regulated by various factors, such as temperature and the balance of exogenous or endogenous hormones [26]. As Tang [27] et al. reported, that low temperature is the key factor in seed germination of *T. iliensis*. Rouhi [28] et al. also found that the seed dormancy of *T. kaufmanniana* could be broken after cold stratification for 49 days. Abscisic acid (ABA) and gibberellin (GA) play a central role in seed dormancy regulation [29]. Gibberellin was regarded as a positive regulator of seed germination, and the application of exogenous GA could reduce the seed dormancy levels of *T. iliensis* and *T. tarbagataica* [30]. Furthermore, in a recent study, we also showed that the addition of gibberellin increased the seed germination in *T. thianschanica* [31]. However, the response of wild tulip species to temperature is different, especially for some native and rare wild resources that have not seen a report yet. The germination stage is a crucial one in a plant's life cycle, and seed dormancy prevents the emergence of seedlings of wild species in the wrong season or place [32,33]. Our previous study found that under natural conditions, the wild tulips had low germination—less than 50%—and the seedlings were easily affected by the environment. Therefore, the study of the seed dormancy mechanism and germination behavior of wild tulips is important to promote their propagation ex situ.

To date, a lot of valuable research has been carried out on the taxonomy, phenology, and anatomy of *Tulipa* species. However, seeds are a vital part of the sexual propagation and population establishment of this genus, but there are only a few reports on their morphology and germination characteristics. The present study compared seed morphological features among the wild tulip species and examined the potential value of these characteristics in systematics. In addition, several treatments were used to break seed dormancy and study if there is a relationship between seed morphology and seed germination characteristics in the genus *Tulipa*.

## 2. Materials and Methods

### 2.1. Seed Collection

Seeds of eight wild tulip species were collected in May-June 2018 from natural populations in Xinjiang Provinces, China (Table 1). For each species, 150 capsules with mature seeds were randomly collected. After natural drying in the room, seeds were peeled from the capsules. Nonseed structures and empty seeds were removed by hand, and brown seeds with visible embryos were selected out and stored in paper bags at room temperature for the germination experiment.

**Table 1.** Information of the eight wild *Tulipa* species native to China.

| Sections | Species | Seed Sources | Latitude and Longitude (N, E) | Altitude (m) | No. of Accessions |
|---|---|---|---|---|---|
| *Amana* | *T. edulis* | Dalian city, LP | 38°57′, 121°27′ | 50 | 4 |
| *Eriostemones* | *T. patens* | Bole city, XP | 44°87′, 82°35′ | 692 | 4 |
| | *T. buhseana* | Tuoli city, XP | 46°11′, 83°36′ | 635 | 8 |
| *Leiostemones* | *T. altaica* | Tacheng city, XP | 47°01′, 83°12′ | 1147 | 5 |
| | *T. thianschanica* | Zhaosu city, XP | 43°09′, 81°10′ | 1852 | 10 |
| | *T. sinkiangensis* | Urumchi city, XP | 43°55′, 87°49′ | 937 | 6 |
| | *T. iliensis* | Yamalic city, XP | 43°47′, 87°33′ | 1080 | 6 |
| | *T. schrenkii* | Yumin city, XP | 46°12′, 83°02′ | 715 | 5 |

Note: LP—Liaoning province; XP—Xinjiang Province.

### 2.2. Seed Morphological Characteristics Observation

Seed size and embryo length were measured with three replicates of 90 seeds using the Olympus SZX7 stereomicroscope. Seed colors were observed according to the Royal Horticultural Society (2010) Color Charts (RHSCC). Seed coat features were observed by SEM (TM3030, Hitachi, Japan). Before observation, the seed surface was cleaned softly with a brush and then fixed on an object stage. The samples were sputter-coated with a gold layer (MSP-1S, Shinkku VD, Japan), and photographed at 15 KV. The terminology of ornamentation was mainly followed [9]. In order to simultaneously use quantitative and qualitative variables, the UPGMA algorithm and Gower's similarity index were used in cluster analysis. The eight characters or variables, including quantitative and qualitative attributes of seed morphology, were defined to perform cluster analysis on eight wild tulip species (Table 2).

**Table 2.** The seed characters used in cluster analysis.

| Characters | Unit or Code |
|---|---|
| Seed size (Seed length × Seed width) | mm |
| E/S (Embryo length/Seed length) | ratio |
| Seed shape | 0 = half-moon; 1 = sector |
| Seed color | 0 = deep-brown; 1 = brown; 2 = light-brown |
| Embryo morphology | 0 = linear, invisible; 1 = linear, visible |
| Testa surface | 0 = stripes; 1 = reticulate |
| Type of murus | 0 = convex; 1 = concave; 2 = flat |
| Type of meshes | 0 = smooth; 1 = granular; 2 = verrucate; 3 = powdery |

### 2.3. Seed Germination Test

The germination tests were performed simultaneously on the eight wild tulip species using three replicates of 100 seeds per treatment. All germination experiments were conducted in incubators lined with two layers of wet filter paper placed in climatic chambers RUMED 1301 (Rubarth Apparate GmbH, Germany). To maintain the solution at a relatively constant level, distillate water was added as necessary to supplement the evaporation loss. Seeds were considered to have germinated when the visible radicle measured > 1.5 mm.

The germination percentage (GR) and the time add up to 50 percent of the seed germination ($T_{50}$), which were calculated as the average of three replications. The $T_{50}$ was calculated per treatment according to the following equation:

$$T_{50} = di + ((N/2 - ni)(dj - di))/(nj - ni) \tag{1}$$

T5

where N is the final number of germinated seeds, ni and nj (ni < N/2 < nj) are the total number of germinated seeds by adjacent counts at time di and dj, respectively [34].

2.3.1. Effect of Seed Morphology and Temperatures on the Germination

To determine the effect of seed morphology and temperature on germination, seeds were grown in an incubator with constant and fluctuating temperatures. Seed size, embryo length, E/S, and seed color were tested before germination experiment. Twenty seeds were randomly selected every 5 days for the determination of seed morphology, and measured by an Olympus SZX7 stereomicroscope. Seed colors were observed according to RHSCC. Three constant temperature treatments were carried out at 4, 10, and 16 °C in the dark, and three alternating temperature treatments were conducted in the dark as follows: (A) 4/10 °C; (B) 10/16 °C; (C) 4/16 °C. Germination rate was counted every three days, and seeds were considered to have completed germination when no more seeds germinated for 20 consecutive days.

2.3.2. Effect of Seed Coat on the Germination

Two germination tests were conducted on the seed germination before constant 4 °C and fluctuating 4/16 °C stratification; one type was the seeds with intact seed coats and another type was the seeds without seed coats.

2.3.3. Effects of Gibberellic Acid (GA$_3$) and Paclobutrazol (PAC) on the Germination

The fresh intact seeds were incubated in aqueous solutions containing 100, 200, and 400 μM GA$_3$ and its inhibitor, paclobutrazol (PAC), respectively, for 24 h, and the seeds were then subjected to the germination test at 4 °C and 4/16 °C, the temperature test (Section 2.3.1) without GA as a direct control.

*2.4. Statistical Analyses*

Statistical analyses were performed using SPSS 23.0 (IBM, Armonk, NY, USA) software at a significant level of $p < 0.05$. Means data were compared by Duncan's multiple range tests to determine significant differences among treatments. Values were expressed as the mean $\pm$ SE of three replicates in each of the independent experiments, and all figures were produced using GraphPad Prism 8.0.1 for Windows (GraphPad Software, San Diego, CA, USA).

**3. Results**

*3.1. Seed Morphology of Tulipa Species*

There were, to some extent, differences in seed morphology and size among the eight wild *Tulipa* species. The first was seed shape; the *T. edulis* seed was half-moon-shaped with the embryo invisible, but those of the other seven *Tulipa* species were sector shaped with linear and visible embryos (Figure 1). The second was seed size (Table 3), the largest seed dimension was 7.36 mm × 6.21 mm in *T. altaica* and the smallest seed dimension was 4.53 mm × 3.54 mm in *T. iliensis*. The ratio of embryotic length to seed length (E/S) was less than 0.45 in *T. edulis, T. altaica, T. patens, T. thianschanica, T. sinkiangensis,* and *T. iliensis,* while they were up to 0.62 and 0.59 in *T. buhseana* and *T. schrenkii*, respectively. The third seed colors were divided into three categories according to the Royal Horticultural Society color charts.

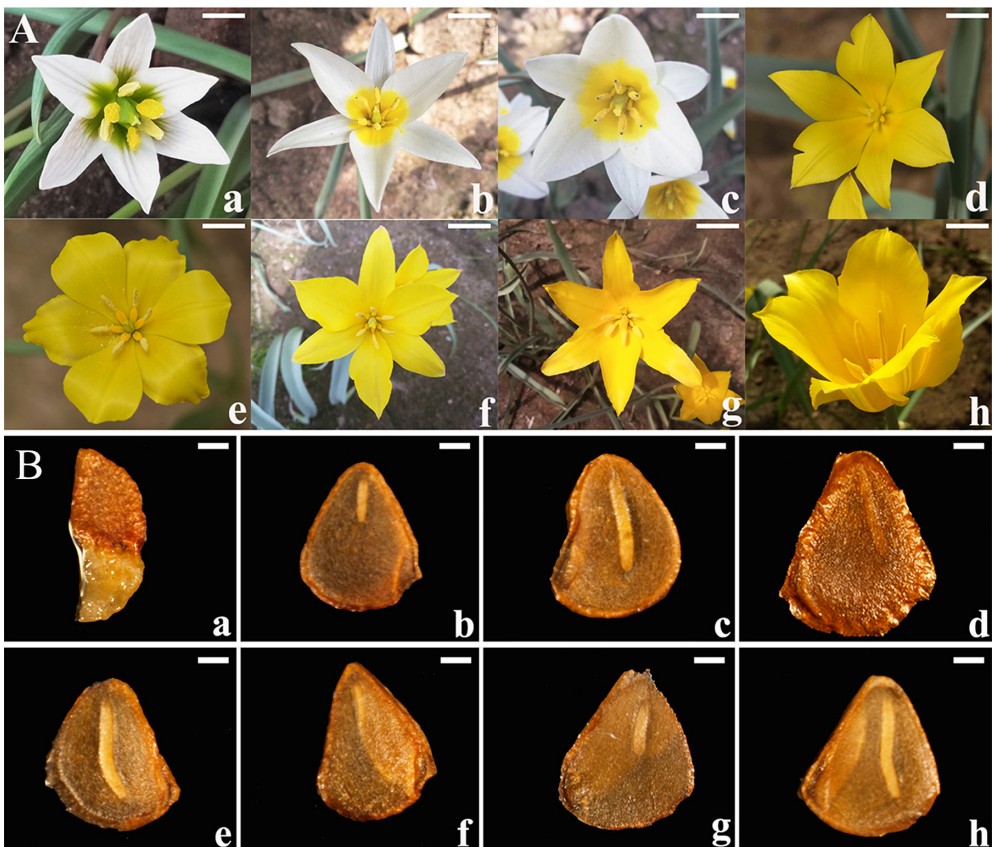

**Figure 1.** The flowers and seeds of eight wild *Tulipa* species native to China. Bar = 1 cm (**A**). flowers (**B**). seeds; (**a**) to (**h**) in turn represent respectively *T. edulis*, *T. patens*, *T. buhseana*, *T. altaica*, *T. thianschanica*, *T. sinkiangensis*, *T. iliensis*, and *T. schrenkii*.

**Table 3.** The seed characters of the eight wild *Tulipa* species native to China.

| Species | Seed Size (mm) | Embryo Length (mm) | E/S | Seed Color | Seed Shape | Embryo Morphology | Seed Coat Surface | Type of Murus | Type of Meshes |
|---|---|---|---|---|---|---|---|---|---|
| *T. edulis* | 5.86 × 2.43 | 1.12 ± 0.05f | 0.19 ± 0.01e | Deep-brown (175A) | Half-moon | Linear, invisible | Stripes | Concave | Smooth |
| *T. patens* | 6.42 × 5.17 | 2.52 ± 0.19cd | 0.40 ± 0.04bc | Brown (N170A) | Sector | Linear, visible | Reticulate | Convex | Powdery |
| *T. buhseana* | 5.52 × 4.30 | 3.44 ± 0.17a | 0.62 ± 0.01a | Brown (N170A) | Sector | Linear, visible | Reticulate | Concave | Smooth |
| *T. altaica* | 7.36 × 6.21 | 3.20 ± 0.19ab | 0.44 ± 0.03bc | Deep-brown (175B) | Sector | Linear, visible | Reticulate | Concave | Powdery |
| *T. thianschanica* | 4.89 × 3.86 | 2.08 ± 0.09de | 0.43 ± 0.01bc | Light-brown (164A) | Sector | Linear, visible | Reticulate | Convex | Verrucate |
| *T. sinkiangensis* | 6.29 × 5.13 | 2.85 ± 0.23bc | 0.45 ± 0.03b | Brown (N170B) | Sector | Linear, visible | Reticulate | Convex | Granular |
| *T. iliensis* | 4.53 × 3.54 | 1.65 ± 0.12e | 0.37 ± 0.02d | Light-brown (165B) | Sector | Linear, visible | Reticulate | Flat | Smooth |
| *T. schrenkii* | 5.43 × 3.79 | 3.18 ± 0.09ab | 0.59 ± 0.02a | Light-brown (166D) | Sector | Linear, visible | Reticulate | Flat | Smooth |

Note: Data are the mean ± SE, different lower-case letters within the same column indicate a significant difference at $p < 0.05$ with Duncan's multiple range test.

### 3.2. Seed Morphology Cluster Analysis of Tulipa Species

According to the seed coat micromorphology observed by SEM (Figure 2), eight *Tulipa* species could be divided into two types, the first type included only one species, *T. edulis*, which has irregular and longitudinally arranged stripes ornamentation (Figure 2A), and the second type included the other seven species, all of them shared the same irregular reticulate ornamentation (Table 3). From Table 3, except for *T. iliensis* and *T. schrenkii* that had flat muri, the rest of the species all had concave muri. The types of meshes were also

different, including smooth, powdery, verrucate, and granular. According to the type of murus and mesh, four subclasses were further divided (Figure 2). Subclass I included *T. patens* and *T. altaica* (Figure 2b,c). Subclass II included *T. thianschanica* and *T. sinkiangensis* (Figure 2d,e). Subclass III included *T. iliensis* and *T. schrenkii* protruding and wide muri and flat and wide meshes were their typical characteristics (Figure 2f,g). Subclass IV only included *T. buhseana*, which had deep and smooth meshes (Figure 2h).

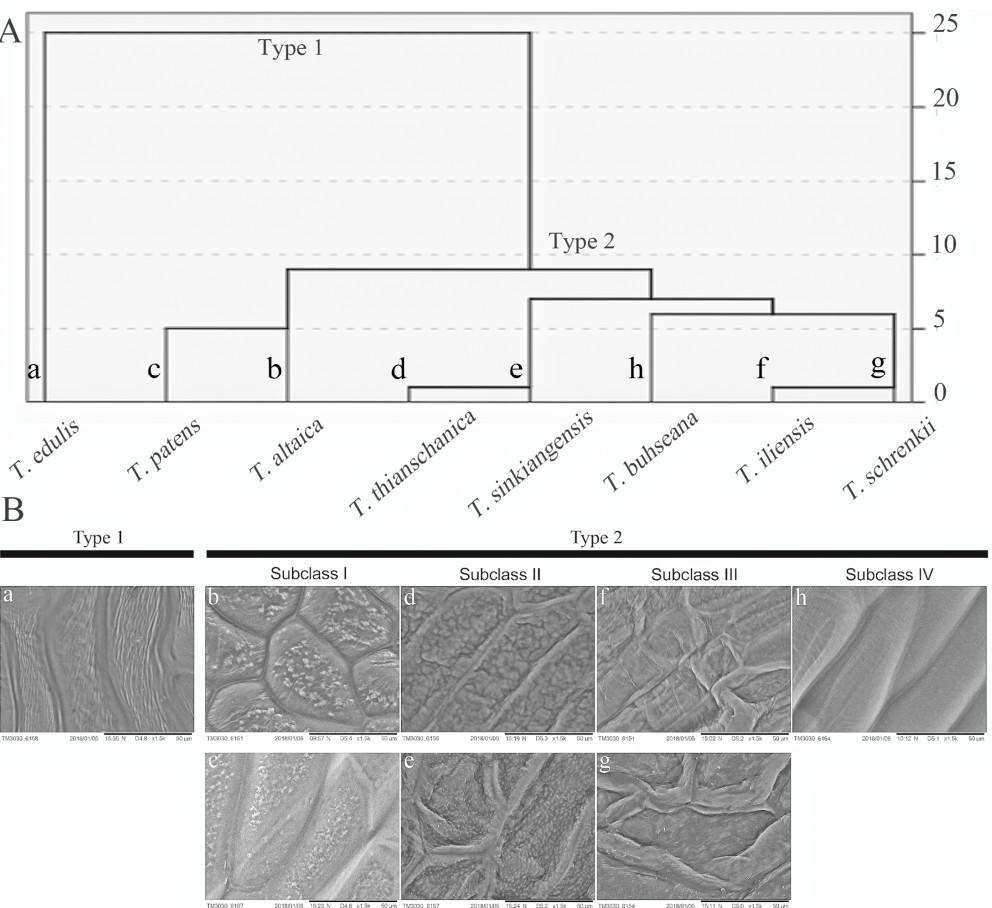

**Figure 2.** The cluster analysis of eight wild *Tulipa* species native to China. Bar = 50 μm. (**A**). the cluster results (**B**). seed coat micromorphology observed by SEM; (**a**) to (**h**) in turn represent respectively *T. edulis*, *T. patens*, *T. buhseana*, *T. altaica*, *T. thianschanica*, *T. sinkiangensis*, *T. iliensis*, and *T. schrenkii*.

### 3.3. Effect of Seed Morphology and Temperatures on Germination

Both constant and fluctuating temperatures significantly affected the germination rate of eight wild tulip species (Figure 3). For *T. edulis*, the highest germination rate was 38.7% at 10 °C, and the remaining treatments were lower than 35%. In the other seven species, germination was only approximately 1.3% at 16 °C and/or 10/16 °C, seed germination increased to greater than 50% (>50%) at 4 °C and 4/16 °C. From Figure 3, the germination rate increased with large fluctuations in temperature, such as *T. patens*, where the germination rate was 48.5% at 4/10 °C, but increased to 77.8% at 4/16 °C. Slightly different from *T. sinkiangensis* and *T. schrenkii*, their germination rates were 58.3% and 59.1% at 4/16 °C, and this germination rate was less than 45% at 4 °C, the others showed satisfactory germination rates at 4 °C. As a whole, either 4 °C or 4/16 °C was the most suitable temperature for these seven-tulip seed germination.

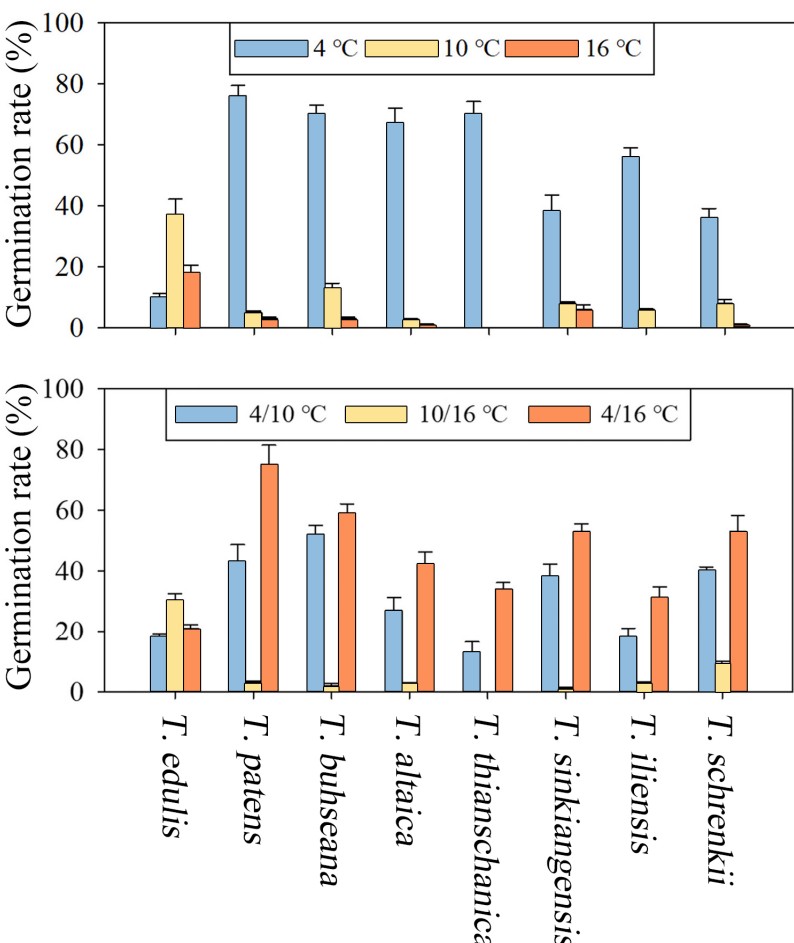

**Figure 3.** Effects of temperatures on seed germination of eight wild *Tulipa* species native to China.

As shown in Table 4, these results showed that seed size was significantly correlated with embryotic length and E/S, with correlation coefficients of 0.448 and 0.298, respectively, but had no correlation with seed color. A high positive correlation was obtained between the seed morphology and temperatures on germination ($T_{50}$). The embryotic length, E/S, and seed coat color were significantly correlated with germination at $p < 0.05$ level, seed size, and temperature were significantly correlated with germination at $p < 0.01$ level.

**Table 4.** Correlations between temperatures and seed morphological indexes on germination.

| Indexes | Seed Size | Embryo Length | E/S | Seed Color | Temperatures | Germination ($T_{50}$) |
|---|---|---|---|---|---|---|
| Seed size | 1 | | | | | |
| Embryo length | 0.448 * | 1 | | | | |
| E/S | 0.298 | 0.926 ** | 1 | | | |
| Seed color | 0.133 | 0.218 | 0.102 | 1 | | |
| Temperatures | 0.035 | 0.612 ** | 0.545 ** | 0.743 ** | 1 | |
| Germination ($T_{50}$) | 0.795 ** | 0.683 ** | 0.842 ** | 0.472 * | 0.944 ** | 1 |

Note: Significant differences indicated * $p < 0.05$, ** $p < 0.01$.

### 3.4. Effect of Seed Coat on Germination

There was little effect on seed germination ($T_{50}$) at the optimum temperatures of 4 °C and/or 4/16 °C when seed coats were removed (Table 5). For WSC treatment, the $T_{50}$ of *T. edulis* and *T. altaica* was significantly faster than ISC at 4 °C, especially for *T. altaica*, approximately 20 days will be shortened at 4/16 °C(86.74, 66.72). Compared with 4 °C

treatment, the $T_{50}$ of *T. patens*, *T. buhseana*, and *T. iliensis* benefited more from WSC at a fluctuating temperature of 4/16 °C, with values of 53.70 to 44.12, 47.36 to 35.45, and 57.97 to 50.30, respectively.

**Table 5.** Effects of seed coat on the $T_{50}$ of the eight wild *Tulipa* species native to China.

| Species | 4 °C, ISC | 4 °C, WSC | 4/16 °C, ISC | 4/16 °C, WSC |
|---|---|---|---|---|
| *T. edulis* | 105.10 ± 8.17 a | 95.58 ± 6.88 b | 84.35 ± 8.50 c | 75.18 ± 6.02 d |
| *T. patens* | 38.83 ± 2.12 c | 36.50 ± 1.23 c | 53.70 ± 2.19 a | 44.12 ± 3.58 b |
| *T. buhseana* | 29.75 ± 4.13 c | 27.26 ± 3.21 c | 47.36 ± 0.83 a | 35.45 ± 3.14 b |
| *T. altaica* | 57.27 ± 1.95 c | 49.33 ± 5.21 d | 86.74 ± 3.32 a | 66.72 ± 5.44 b |
| *T. thianschanica* | 48.13 ± 3.01 b | 49.50 ± 5.08 b | 79.10 ± 3.01 a | 78.53 ± 6.31 a |
| *T. sinkiangensis* | 82.83 ± 8.45 a | 74.37 ± 7.27 a | 44.40 ± 1.37 b | 41.22 ± 5.24 b |
| *T. iliensis* | 39.50 ± 1.15 c | 37.97 ± 5.30 c | 57.97 ± 1.50 a | 50.30 ± 2.11 b |
| *T. schrenkii* | 65.10 ± 1.87 a | 69.53 ± 4.55 a | 52.43 ± 0.85 b | 48.30 ± 3.32 b |

Notes: ISC, iNotes: ISC, intact seed coat; WSC, without seed coat. Data are the mean ± SE, different lower-case letters within the same column indicate a significant difference at *p* < 0.05 with Duncan's multiple range test.

*3.5. Effect of GA$_3$ and PAC on Germination*

The responses of seed germination to GA$_3$ and PAC were different at 4 °C and/or 4/16 °C. As shown in Table 6, *T. buhseana* has the shortest $T_{50}$ (29.75 d), followed by *T. patens* (38.83 d) and *T. iliensis* (39.50 d), respectively, and $T_{50}$ of the rest five *Tulipa* species were longer than 50 days in control. The results showed exogenous gibberellin promoted seed germination and its inhibitor PAC delayed seed germination in all test species, and the effect of GA$_3$ varied greatly among species and also depended on its concentration at 4 °C. Such as, in *T. sinkiangensis*, adding 100 µM GA$_3$ ($T_{50}$, 58.10 d) made seeds germinate significantly earlier than control ($T_{50}$, 82.83 d), while in *T. iliensis*, the $T_{50}$ value (38.30 d) was similar to that of control (39.50 d). At 4/16 °C, *T. buhseana* and *T. sinkiangensis* had the shortest $T_{50}$ (44.40 d and 47.36 d, respectively), followed by *T. patens* (53.70 d), *T. iliensis* (57.97 d), *T. schrenkii* (52.43 d), and $T_{50}$ of the other three species was longer than 80 days. Exogenous gibberellin did not affect the $T_{50}$ of *T. sinkiangensis* and *T. schrenkii*, but significantly promoted seed germination of the other six *Tulipa* species depending on its concentration. 100 µM GA$_3$ was considered the most effective concentration for tulip seed germination, and it had a negative effect on germination when its concentration increased above 200 µM (Table 6).

**Table 6.** Effects of GA$_3$ and its inhibitor PAC on the $T_{50}$ of the eight wild *Tulipa* species native to China.

| | Treatments (µM) | | *T. edulis* | *T. patens* | *T. buhseana* | *T. altaica* | *T. thianschanica* | *T. sinkiangensis* | *T. iliensis* | *T. schrenkii* |
|---|---|---|---|---|---|---|---|---|---|---|
| 4 °C | | CK | 105.10 ± 8.17 b | 38.83 ± 2.12 d | 29.75 ± 4.13 b | 57.27 ± 1.95 b | 48.13 ± 3.01 d | 82.83 ± 8.45 a | 39.50 ± 1.15 c | 65.10 ± 1.87 a |
| | GA$_3$ | 100 | 92.38 ± 7.50 c | 42.17 ± 2.10 c | 24.30 ± 1.80 c | 52.77 ± 1.66 c | 38.67 ± 5.86 e | 58.10 ± 2.88 cd | 45.92 ± 1.25 b | 52.83 ± 1.50 d |
| | | 200 | 86.35 ± 9.13 cd | 36.17 ± 2.78 e | 25.37 ± 1.21 c | 56.37 ± 0.85 b | 45.83 ± 4.01 d | 62.00 ± 4.07 c | 45.30 ± 0.79 b | 56.83 ± 1.39 c |
| | | 400 | 106.22 ± 5.92 b | 43.40 ± 0.98 c | 26.33 ± 0.81 c | 57.63 ± 1.96 b | 54.60 ± 4.93 c | 66.70 ± 4.96 b | 38.30 ± 1.05 c | 60.12 ± 1.95 b |
| | PAC | 100 | 144.66 ± 3.59 a | 75.00 ± 2.95 b | 57.33 ± 1.53 a | 61.97 ± 1.39 a | 68.33 ± 2.52 b | 87.92 ± 5.11 a | 71.07 ± 2.76 a | 62.46 ± 0.95 a |
| | | 200 | 156.37 ± 8.50 a | 77.60 ± 4.16 b | 58.67 ± 4.51 a | 67.50 ± 1.35 a | 73.66 ± 4.16 b | 85.63 ± 3.47 a | 74.37 ± 2.16 a | 64.73 ± 5.80 a |
| | | 400 | 148.22 ± 9.20 a | 81.10 ± 9.41 a | 60.60 ± 1.31 a | 64.33 ± 2.75 a | 81.33 ± 5.51 a | 84.10 ± 4.37 a | 72.77 ± 2.55 a | 65.20 ± 0.92 a |
| 4/16 °C | | CK | 84.35 ± 8.50 b | 53.70 ± 2.19 c | 47.36 ± 0.83 b | 86.74 ± 3.32 b | 79.10 ± 3.01 c | 44.40 ± 1.37 c | 57.97 ± 1.50 c | 52.43 ± 0.85 b |
| | GA$_3$ | 100 | 69.38 ± 5.39 d | 61.40 ± 2.15 b | 33.50 ± 2.29 c | 65.00 ± 2.42 c | 74.67 ± 8.96 cd | 44.03 ± 4.01 c | 53.07 ± 0.96 c | 50.83 ± 9.35 b |

**Table 6.** *Cont.*

| Treatments (μM) | | *T. edulis* | *T. patens* | *T. buhseana* | *T. altaica* | *T. thianschanica* | *T. sinkiangensis* | *T. iliensis* | *T. schrenkii* |
|---|---|---|---|---|---|---|---|---|---|
| | 200 | 75.67 ± 9.52 c | 63.27 ± 1.90 b | 33.00 ± 1.00 c | 65.27 ± 2.15 c | 81.33 ± 3.79 c | 43.55 ± 1.03 cd | 53.10 ± 1.95 c | 45.60 ± 2.95 c |
| | 400 | 82.85 ± 7.31 b | 64.10 ± 2.72 b | 35.23 ± 1.30 c | 68.97 ± 3.18 c | 94.33 ± 6.51 b | 46.03 ± 1.05 b | 52.43 ± 0.71 c | 50.06 ± 3.10 b |
| PAC | 100 | 124.40 ± 7.11 a | 113.00 ± 10.16 a | 57.83 ± 5.42 a | 104.53 ± 6.85 a | 99.67 ± 9.71 b | 47.90 ± 1.35 b | 108.53 ± 5.53 b | 68.40 ± 1.71a |
| | 200 | 132.47 ± 6.39 a | 100.67 ± 9.45 a | 57.53 ± 2.32 a | 103.33 ± 6.14 a | 140.32 ± 3.51 a | 46.43 ± 0.71 bc | 130.10 ± 1.75 a | 62.47 ± 3.91 a |
| | 400 | 128.45 ± 6.44 a | 95.13 ± 8.95 a | 59.13 ± 1.75 a | 105.83 ± 3.93 a | 135.67 ± 7.09 a | 51.80 ± 1.37 a | 123.33 ± 2.65 a | 66.67 ± 4.66 a |

The data are the mean ± SE, different lower-case letters within the same column indicate a significant difference at $p < 0.05$ with Duncan's multiple range test.

## 4. Discussion

### 4.1. Seed Morphology and its Taxonomic Significance for Eight Wild Tulip Species

From our results, the seed of *T. edulis* had a half-moon shape with an obvious seed stalk, an invisible embryo, and an irregular and/or longitudinal stripe on the testa. However, the seeds of the other seven species shared a similar sector seed shape, reticulate ornamentation, and rectangular or irregular meshes on the testa. This showed that there were significant differences in seed morphology and micromorphology between *T. edulis* and the core groups of *Tulipa*, such as Sect. *Eriostemons* and Sect. *Leiostemons* [35]. Therefore, our result supported the viewpoint that the *Amana* group should be isolated from the tulip genus and restored its status as an independent *Amana* genus. The response to temperature and the difference in germination characteristics between *T. edulis* and the other seven tulip species also supported the above conclusion.

Sect. *Leiostemones* is the largest group in the genus *Tulipa*, with 69 species distributed in Central Asia, West Asia, Europe, and the Mediterranean. The interspecific relationship in this section is more complex because of the diversity of species and morphological variation. Among the seven species distributed in Xinjiang, China. Vvedenskii [36] thought that *T. schrenkii*, *T. altaica*, and *T. iliensis* had a close genetic relationship. Hall [37] observed the morphological characteristics of the seed coat and concluded that *T. iliensis*, *T. sinkiangensis*, and *T. schrenkii* had a reticular pattern with a clear outline and deep meshes, which may have a closer phylogenetic relationship. In addition, we also found the relationship between *T. patens* and *T. buhseana* might be far away based on testa micromorphology, which was consistent with the result of Botschantzeva [38]. Further research will combine the evidence of leaf epidermis, pollen, and molecular systematics to make a comprehensive analysis of this section and reveal the complex interspecific relationship.

### 4.2. Seed Germination Characteristics of Eight Wild Tulip Species

The tulip seeds had underdeveloped embryos. The fresh seeds of eight tulip species had an E/S ratio of approximately 0.5, which increased to 1.0 before radicle emergence, regardless of any treatments. Morphological dormancy (MD) was defined as the presence of incomplete embryotic development in the seed that could germinate within 30 days under the appropriate environmental conditions [39]. However, in our experiment, the seeds of wild tulips without low temperature stratification could not germinate. Therefore, we concluded that the seeds of *Tulipa* had morphophysiological dormancy (MPD). According to Baskin and Baskin [40], who proposed nine types of MPD, which can be subdivided into two categories: simple and complex, based on temperature requirements for embryonic growth, warm and/or cold stratification for germination, and responses of seeds to gibberellic acid ($GA_3$) pretreatment. In the simple levels of MPD, embryos grew at warm temperatures (≥15 °C), and the physiological dormancy (PD) was broken prior to embryotic elongation. On the other hand, low temperatures (0–10 °C) were required for embryotic elongation in the complex levels of MPD [39]. Our results showed that the treatment from a lower temperature (4 °C) to a higher temperature (16 °C) can improve the embryo growth of

tulip seeds, but these species seeds could not germinate if they were treated at more than 16 °C. Therefore, we concluded that the seeds of tulips had a complex type of MPD. Further, we also tried to classify the level of complex MPD, which included three levels, nondeep, intermediate, and deep [41,42]. In our study, gibberellin (GA$_3$) pretreatment was positive for promoting the germination of tulip seeds, depending on its applied concentration. Therefore, the seed of tulips had a nondeep complex MPD.

Seed germination is a key life-history stage. Whether a seed germinates or not is critical in the regeneration of populations, especially endangered species [43]. Normally, wild tulip seed has low germination in situ owing to an unfavorable natural environment and an undeveloped seed. Previously, reports showed that seed morphology was closely related to seed germination [44]. In our results, seed size was significantly correlated with embryo length and E/S but had no correlation with seed color. A high positive correlation was obtained between seed morphology and temperatures at germination ($T_{50}$). Embryo length, E/S, and seed coat color were significantly correlated with germination; seed size and temperature were significantly correlated with germination at $p < 0.01$ level, which is in accordance with Gholami [45], who observed an increase in germination as well as a greater speed of germination in larger seeds compared with small seeds in the common bean (*Phaseolus vulgaris* L.). More attention needs to be paid to the relationship between the content of nutrients in tulip seeds and dormancy and/or germination.

Temperature experiments on eight wild tulips showed that there was a high germination percentage at a constant 4 °C and at a fluctuating temperature 4/16 °C, while germination was progressively inhibited at a constant 16 °C or higher, except for *T. edulis*, for which the most suitable temperature for germination was 10 °C. These results showed that tulip seeds exhibited a narrow thermal range for germination, which explained why those species were dormant in summer and germinated in spring. The ability to germinate at 4 °C in darkness suggested that seeds would germinate even under the cover of snow. This information could be used to design constant-cold temperature practices to manage tulip populations.

To our knowledge, there was no recommendation for a specific concentration of hormones to break the seed dormancy of tulips. In this study, we found that under 4 °C and 100 μM GA$_3$, the germination of *T. patens*, *T. buhseana*, *T. thianschanica*, and *T. sinkiangensis* significantly increased. Similarly, gibberellin promoted seed germination of *T. buhseana* and *T. schrenkii* at fluctuating temperatures (100 μM, 4/16 °C). However, the negative effect would be produced on those tulip species when the concentration of GA$_3$ was over 200 μM. Gibberellin was known to stimulate various other enzymes in the early stages of seed germination [46,47]. Moreover, it has been previously reported that external gibberellin absorbed by the seed reduced the concentration of ABA and accelerated the catabolism of stored nutrients [47,48]. In our study, although the effect of exogenous GA$_3$ and PAC on the embryonic length was insignificant before the radicle broke through the seed coat, they significantly promoted and/or inhibited seed germination time at any concentration. Therefore, we believed that the balance of gibberellin content between external and internal decided the start, maintenance, and end of seed dormancy of these tulip species, and the changes of endogenous hormones and their response to exogenous hormones in tulips will be determined to prove our hypothesis. As a whole, GA$_3$ promoted the germination of tulip seeds, either at a constant temperature of 4 °C or a fluctuating temperature of 4/16 °C, but it could not replace the low temperature. Future research should pay more attention to the interaction between hormones and other treatment methods to accelerate the breaking of seed dormancy.

## 5. Conclusions

The morphology of the tulip seed, especially its ornamentation on the seed coat, had great value in the taxonomy of the genus *Tulipa*. From our results, *T. edulis*, with its half-moon-shaped seed and stripe ornamentation, was significantly different from the other seven species with sector-shaped seeds and reticulate ornamentation. Therefore, we

support the notion that *T. edulis* should be regarded as an independent genus—*Amana*. Additionally, four types of seed surfaces were revealed by the SEM. We also found that the mesh, muri, and surface features of the seed coat contributed to the taxonomy at the section level. On the other hand, fresh seeds of eight tulip species possess a type of nondeep complex MPD. The embryo could grow at 4 °C and 4/16 °C, but seeds could not germinate until after more than two weeks of cold stratification. Under natural conditions, it took 7–9 months from maturity to germination for wild tulip seeds. Our study provided a way for the seed germination to be shortened within 2 months under the experimental conditions. The knowledge acquired in this work for both germination and emergence of *Tulipa* seeds will help establish management programs for utilizing and conserving these wild resources.

**Author Contributions:** Conceptualization, J.L. and W.Z.; methodology, W.Z. and J.Z.; validation, W.Z., J.Z. and L.X.; formal analysis, J.L. and W.Z.; investigation, W.Z.; resources, J.L.; data curation, H.D.; writing—original draft preparation, W.Z.; writing—review and editing, J.L., W.Z., L.X. and H.D.; visualization, J.L. and W.Z.; supervision, H.D.; project administration, J.L.; funding acquisition, J.L., W.Z. and J.Z. All authors have read and agreed to the published version of the manuscript.

**Funding:** This study was financially supported by the Youth Scientific Research Project of Shanxi Province (No. 20210302124023), the Science and Technology Innovation Project of Higher Education Institutions in Shanxi Province (No. 2021L117), the Doctoral Research Initiation Project of Shanxi Agricultural University (No. 2021BQ33), and the National Science Foundation of China (No. 31902044).

**Institutional Review Board Statement:** Not applicable.

**Data Availability Statement:** The data used in this study cannot be shared at present.

**Conflicts of Interest:** The authors declare no conflict of interest.

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
