# Peer review of "Seed Morphology and Germination of Native Tulipa Species"

_agriculture, doi:10.3390/agriculture13020466_

Round 1
Reviewer 1 Report
Dear Authors
- in Introduction, you should reference more studies about the relationship between seed germination and seed morphology. I think the novelty of this work is the relation between seed morphology and seed germination rate and percentage.
- in material and methods, the section; the effect of seed morphology and temperature on seed germination, it is unclear. Please explain some more in this section
-in material and method, and results, correct "effect of GA3 and PAC.."
- Figure 4 is not clear, please change it or remove it.
- Table 6, column 1 from left to right, is not clear. please change its title....you can not write, species and temperature on the same title. Please rewrite this table again,
- in need to discuss more about the role of seed morphology and seed germination,...
Author Response
Dear Review:
Thanks for your valuable advice, which is great helpful to improve the quality of my manuscript. I hope the revised version can be approved by you.

Reviewer 2 Report
The authors have addressed an informative topic; “Taxonomic significance of seed morphology in Tulipa species and its relationship to seed germination”. The outcomes of the study are very valuable and useful for researcher community. In current form, I am not more positive for publication of the manuscript at this stage. However, some concerns are highlighted that may be addressed to improve the quality of manuscript.
- The title does not look meaningful. Please modify and make it catchier.
- There are no information regarding experimental design and treatment plan.
- Quantitatively conclude all your results at the end of the abstract.
- The introduction section is too short, add some data related to your treatments and the importance of your study. It looks very poor.
- Arrange your introduction section in a systematic scientific way and identify the study gap and provide a robust hypothesis. Try to avoid stating general information and be specific.
- Materials and methods section also needs little improvement.
- Improve the reporting language and avoid jargon. Directly state the results. Authors must quantitatively report their results. Make the results section concise and specific.
- Data presentation may be improved particularly figure captions.
- Discussion section may be improved with recent citation. Try to discuss results with recent literature with logical reasoning.
- Quantify the conclusion section as it looks too lengthy.
- Follow the journal guidelines for the references style within the text and in the bibliographic section.
- An extensive improvement in required regarding write up and language.
In the current of manuscript, I am not more positive for publication of the manuscript but it may be accepted for publication after addressing the suggestion and comments.
Author Response

(The authors gave the same response as above.)

Reviewer 3 Report
Dear authors,
Many thanks for sharing an interesting manuscript and description of methods for improving germination and per se propagation of native Tulip species.
I drafted some questions and sugestions, which you are going to find in the attached pdf report.
Best regards,

Author Response

(The authors gave the same response as above.)

Round 2
Reviewer 2 Report
Authors have addressed most of comments and manuscript is significantly improved.